# Physiological basis for atmospheric methane oxidation and methanotrophic growth on air

Tilman Schmider [1] ✉, Anne Grethe Hestnes[1], Julia Brzykcy [2], Hannes Schmidt [3], Arno Schintlmeister [4], Benjamin R. K. Roller [4], Ezequiel Jesús Teran[5,6], Andrea Söllinger[1], Oliver Schmidt [1], Martin F. Polz [4], Andreas Richter [3], Mette M. Svenning[1] & Alexander T. Tveit [1] ✉

Atmospheric methane oxidizing bacteria (atmMOB) constitute the sole biological sink for atmospheric methane. Still, the physiological basis allowing atmMOB to grow on air is not well understood. Here we assess the ability and strategies of seven methanotrophic species to grow with air as sole energy, carbon, and nitrogen source. Four species, including three outside the canonical atmMOB group USCα, enduringly oxidized atmospheric methane, carbon monoxide, and hydrogen during 12 months of growth on air. These four species exhibited distinct substrate preferences implying the existence of multiple metabolic strategies to grow on air. The estimated energy yields of the atmMOB were substantially lower than previously assumed necessary for cellular maintenance in atmMOB and other aerobic microorganisms. Moreover, the atmMOB also covered their nitrogen requirements from air. During growth on air, the atmMOB decreased investments in biosynthesis while increasing investments in trace gas oxidation. Furthermore, we confirm that a high apparent specific affinity for methane is a key characteristic of atmMOB. Our work shows that atmMOB grow on the trace concentrations of methane, carbon monoxide, and hydrogen present in air and outlines the metabolic strategies that enable atmMOB to mitigate greenhouse gases.

During the first two decades after emission to the atmosphere, methane ($CH_4$) is a greenhouse gas 80 times more potent than carbon dioxide ($CO_2$)[1,2]. Since 2007, the atmospheric $CH_4$ concentration (1905 p.p.b.v. in July 2022 https://gml.noaa.gov/ccgg/trends_ch4/), that is responsible for approximately 20% of the direct radiative forcing, has been increasing rapidly[1]. The $CH_4$ increase further accelerated in 2014 and is linked to several causes: A decline in the atmospheric concentration of hydroxyl radicals (OH) which is the main sink of atmospheric $CH_4$, as OH oxidize $CH_4$ in the atmosphere[3–5]; anthropogenic emissions from fossil fuel, agricultural, and waste sources[6]; increased microbial $CH_4$ production in wetlands which suggests that current increases are also driven by feedback responses to global warming[7]. Atmospheric $CH_4$ oxidizing bacteria (atmMOB), a subgroup of aerobic methanotrophs, that oxidize $CH_4$ at its atmospheric trace concentration are the only known biological sink of atmospheric $CH_4$. Compared to the OH sink (~500 Tg), the biological sink is rather small

[1]Department of Arctic and Marine Biology, Faculty of Biosciences, Fisheries and Economics, UiT—The Arctic University of Norway, 9037 Tromsø, Norway. [2]Department of Geomicrobiology, Institute of Microbiology, Faculty of Biology, University of Warsaw, 02-096 Warsaw, Poland. [3]Department of Microbiology and Environmental Systems Science, Division of Terrestrial Ecosystem Research, University of Vienna, 1030 Vienna, Austria. [4]Department of Microbiology and Environmental Systems Science, Division of Microbial Ecology, University of Vienna, 1030 Vienna, Austria. [5]Centro de Investigaciones en Física e Ingeniería del Centro de la Provincia de Buenos Aires (CIFICEN-UNCPBA-CONICET-CICPBA), Pinto 399 Tandil (7000), Argentina. [6]Universidad Nacional del Centro de la Provincia de Buenos Aires, Facultad de Ciencias Exactas, Instituto de Física Arroyo Seco (IFAS), Pinto 399 Tandil (7000), Argentina. ✉e-mail: tilman.schmider@uit.no; alexander.t.tveit@uit.no

as it removes approximately 30 Tg (11–49 Tg) $CH_4$ from the atmosphere every year[4]. However, the biological sink has the potential to grow with increasing $CH_4$ concentrations. This is of particular importance as the decline of atmospheric OH, caused by reaction with atmospheric hydrogen ($H_2$) and other gases, might accelerate due to increasing $H_2$ emissions from a hydrogen-based economy[8,9]. Additionally, the biological sink is within reach of management practices devised to maximize its natural potential and harness it for $CH_4$ removal[10,11]. Yet, substantial uncertainties concerning the size of the biological sink, and the ecology and metabolic basis for growth on atmospheric $CH_4$ by atmMOB, caused by a historical lack of atmMOB in pure culture, has impeded our ability to study, manage, and exploit the sink[10]. In this study, by screening seven methanotrophic species, we have outlined the physiological basis that enables atmMOB to grow on atmospheric $CH_4$ and serve as an atmospheric $CH_4$ sink.

In 1992, ten years after Harriss et al. reported the first indications of atmospheric $CH_4$ oxidation by microorganisms[12], Bender and Conrad concluded from biphasic $CH_4$ oxidation kinetics of soils that an unknown group of methanotrophs might be responsible for atmospheric $CH_4$ oxidation[13]. Since then, two major questions have remained partially unanswered: Which methanotrophs are responsible for oxidation of atmospheric $CH_4$? How can these organisms survive and grow despite the apparent energetic limitations inherent to the oxidation of the low atmospheric $CH_4$ concentrations?

Isotopic labeling studies revealed that members of *Alpha-* and *Gammaproteobacteria* contributed to atmospheric $CH_4$ oxidation and assigned them to the upland soil clusters alpha and gamma (USCα and USCγ)[14–16]. Several environmental and ecological studies have ascribed atmospheric $CH_4$ oxidation mainly to these two clusters[17–20]. However, over the years, studies targeting methanotrophs in upland soils have reported the presence of alphaproteobacterial methanotrophs outside the USCα[14,21–24]. These observations suggest that also conventional methanotrophs (methanotrophs assumed to grow only at high $CH_4$ concentrations), from genera like *Methylocapsa*, *Methylosinus* and *Methylocystis*, could contribute to the atmospheric $CH_4$ sink.

Dunfield[25] summarized three potential lifestyles of atmMOB that might enable cellular maintenance and growth at the low $CH_4$ concentrations in air and the associated energy limitation: (i) Flush feeding on high $CH_4$ concentrations generated periodically in deeper soil layers, in addition to atmospheric $CH_4$ oxidation; (ii) An oligotrophic lifestyle based on atmospheric $CH_4$ as sole carbon and energy source; (iii) A mixotrophic lifestyle to utilize other substrates for energy conservation in addition to $CH_4$.

Flush feeding (i) is supported by the declining potential of methanotrophs to oxidize atmospheric $CH_4$ after several months of $CH_4$ starvation[25,26]. A study on conventional methanotrophs in rice paddy soils showed that methanotrophs regained the ability to oxidize atmospheric $CH_4$ after exposure to high $CH_4$ concentrations[27]. A high specific affinity ($a_A^0$) for $CH_4$ has been suggested as the key trait of oligotrophic atmMOB (ii) to enable growth with air as their only energy and carbon source[25,28]. This assumes that cellular energy requirements for maintenance are 4.5 kJ per carbon mole of biomass per hour (C-mol$^{-1}$ h$^{-1}$) (at 25 °C)[29]. Thus, to survive, oligotrophic atmMOB presumably need an atmospheric $CH_4$ oxidation rate high enough to meet these energy requirements. Such a rate can be achieved by the combination of a high affinity for $CH_4$, reflected in a low half saturation constant ($K_m$), and a high maximum $CH_4$ oxidation rate ($V_{max}$), the fraction of $V_{max}$ and $K_m$ being referred to as $a_A^0$[30]. This theory is supported by low $K_m$ values for $CH_4$ found in several soils[13]. However, in a later study the cell specific $CH_4$ oxidation of USCα members was estimated to be 2.9 to 40 times lower than the presumed rate needed for cellular maintenance[31]. Therefore, the authors considered that a mixotrophic lifestyle (iii) could be the basis for atmospheric $CH_4$ oxidation. In line with this, a recent study

reported the simultaneous oxidation of atmospheric $H_2$, carbon monoxide (CO), and $CH_4$ by *Methylocapsa gorgona* MG08, the first known methanotroph and USCα member in pure culture that can grow on air (with air as sole energy and carbon source)[32,33]. Despite its mixotrophic lifestyle, *M. gorgona* MG08 did not conserve enough energy (0.38 kJ C-mol$^{-1}$ h$^{-1}$) to cover the 2.8 kJ C-mol$^{-1}$ h$^{-1}$ theoretically required to support maintenance at 20 °C (2.8 kJ C-mol$^{-1}$ h$^{-1}$ at 20 °C correspond to 4.5 kJ C-mol$^{-1}$ h$^{-1}$ at 25 °C)[29,32], questioning whether this maintenance energy value and thus a high $a_A^0$ is a relevant benchmark for the physiological capabilities of trace gas oxidizing bacteria. While the initial isolation and study of *M. gorgona* MG08 led to important advancements in our understanding of atmMOB, our knowledge of the metabolic basis allowing atmMOB to grow on atmospheric $CH_4$ remained limited. Recent energy estimates relied on the untested assumptions that all cells remained active over time and could not grow on air without $CH_4$, $H_2$ and CO. Furthermore, these methodologically limited estimates had only been carried out on one strain, *M. gorgona* MG08, and had not been combined with other methods to reveal the metabolic strategies associated to the energy yields.

Here we use filter cultivation to screen six alphaproteobacterial, methanotrophic strains and one gammaproteobacterial, methanotrophic strain for their ability to grow on air. This selection includes the six alphaproteobacterial strains *Methylocapsa gorgona* MG08, *Methylosinus trichosporium* OB3b, *Methylocystis rosea* SV97, *Methylocapsa aurea* KYG, *Methylocapsa acidiphila* B2, and *Methylocapsa palsarum* NE2[33–38] since members of *Methylocapsa*, *Methylosinus* and *Methylocystis* have been observed in net sinks for atmospheric $CH_4$ (upland soils). Additionally, we include *Methylobacter tundripaludum* SV96 as representative methanotroph from the *Gammaproteobacteria*. To outline the physiological basis for growth on air, we perform trace gas oxidation experiments, estimate energy yields from the oxidation of trace gases in air, investigate proteome allocations, and determine the $a_A^0$ for $CH_4$ during growth on air. Based on the lack of knowledge about nitrogen limitation of atmMOB[39], we also evaluate their ability to grow on the nitrogen sources available in air.

## Results and discussion

### Colony formation, trace gas oxidation, and cellular energy yield during growth on air

To test the ability of the seven selected methanotrophs to grow on air, we incubated each strain on filters floating on carbon free medium with ambient air as the sole carbon and energy source, hereafter referred to as filter culture. Microscopy demonstrated that all strains except *Methylobacter tundripaludum* SV96, the gammaproteobacterial strain, formed colonies during six months of incubation (Fig. 1A). To test the metabolic activity of the six months old strains, we performed trace gas oxidation experiments (Fig. 1A, B). During these experiments, filter cultures were floating on mineral medium in bottles with trace concentrations of $CH_4$, $H_2$, and CO in the headspace. The strains, *M. aurea* KYG, *M. gorgona* MG08, *M. palsarum* NE2, and *M. rosea* SV97, hereafter referred to as atmMOB, oxidized $CH_4$, CO, and $H_2$, or $CH_4$ and CO to sub-atmospheric concentrations showing strain-specific oxidation patterns (Fig. 1A, Supplementary Fig. 1). *M. aurea* KYG oxidized CO at the highest rate, followed by $CH_4$ and $H_2$. *M. gorgona* MG08 oxidized $H_2$ at the highest rate, followed by CO and $CH_4$. *M. rosea* SV97 oxidized CO at the highest rate, followed by $CH_4$ and $H_2$. *M. palsarum* NE2 oxidized $CH_4$ and CO at similar rates but did not oxidize $H_2$. The gas oxidation patterns were similar after 12 months of incubation with air (Supplementary Fig. 1). To verify growth on the three trace gases in air, we incubated *M. gorgona* MG08 cells on filters under two different atmospheres: One of synthetic air without the trace gases $CH_4$, CO, and $H_2$ (Gas composition: 400 p.p.m.v. $CO_2$ and 20.9% $O_2$ in $N_2$) and another of ambient air. Growth by colony formation was only observed in the ambient air control, whereas no growth was observed in

synthetic air (Supplementary Fig. 2). This and the repeated observations of trace gas oxidation confirm that these strains can live and grow with air as the sole energy and carbon source. The observed oxidation of at least one atmospheric trace gas in addition to $CH_4$ demonstrates that all four strains are mixotrophic, matching the proposition by Dunfield that mixotrophy could be a physiological basis for atmMOB[25]. Additionally, the strain-specific trace gas uptake patterns demonstrate substantial metabolic differences between these four strains, indicating that multiple metabolic strategies can support oxidation of atmospheric $CH_4$ for growth. These results also demonstrated that the enduring oxidation of atmospheric $CH_4$ after 12 months of growth on

air is not restricted to members of the USCα and USCγ, but also include members from the genera *Methylocystis* and *Methylocapsa*: *M. aurea* KYG, *M. palsarum* NE2, and *M. rosea* SV97 (Fig. 2).

Despite formation of colonies when exposed to air only, *M. acidiphila* B2 and *M. trichosporium* OB3b did not oxidize trace gases after six months of incubation (Fig. 1A). The ability of these two strains to produce polyhydroxyalkanoates (PHA) as storage compounds[35,40] might serve as an explanation for the initial colony formation. Possibly, the two strains accumulated PHA during pre-cultivation at 20% $CH_4$, and then sustained their growth on air by utilizing PHA and atmospheric $CH_4$ as energy and carbon sources until the depletion of their

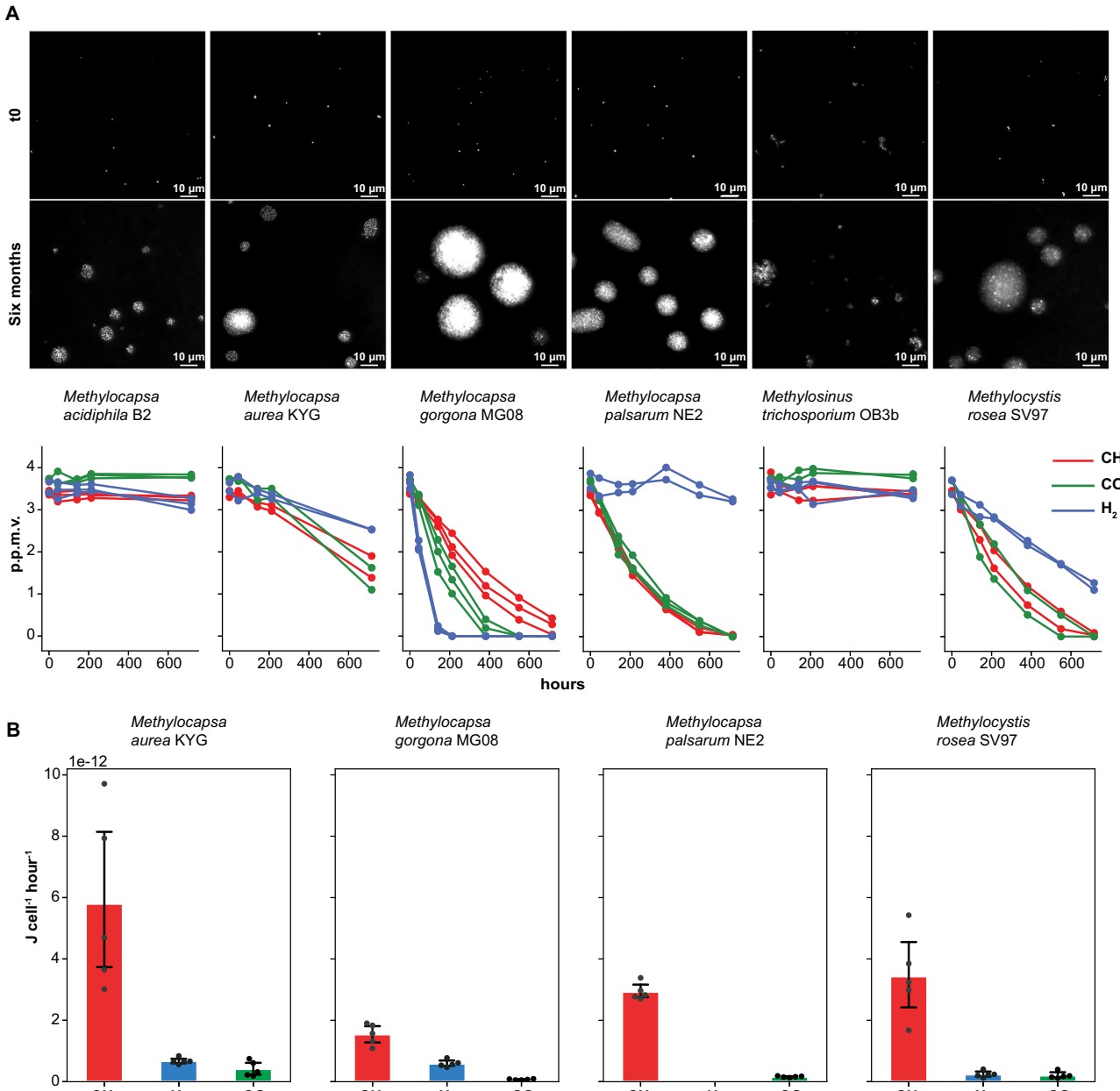

**Fig. 1 | Colony formation and trace gas oxidation on air. A** Images of SYBR green I stained cells and colonies formed by *M. acidiphila* B2, *M. aurea* KYG, *M. gorgona* MG08, *M. palsarum* NE2, *M. trichosporium* OB3b, and *M. rosea* SV97 at incubation start (t0) and after six months of incubation on air as sole energy and carbon source and the related $CH_4$, $H_2$, and CO oxidation at atmospheric pressure displayed in red, blue, and green, respectively. Colony formation experiments with air as sole energy and carbon source have been repeated independently for at least three times per strain all leading to similar results. Source data are provided in the Source Data file

and in the Supplementary Data file (Dataset 1). **B** Cellular energy yield on air. Mean estimated energy yield per cell from the oxidation of atmospheric $CH_4$, $H_2$, and CO displayed in red, blue, and green, respectively, by *M. aurea* KYG, *M. gorgona* MG08, *M. palsarum* NE2, and *M. rosea* SV97. Dots represent the energy yield of the respective biological replicates ($n = 5$) and error bars the standard deviation. Source data are provided in the Source Data file and in the Supplementary Data file (Dataset 3).

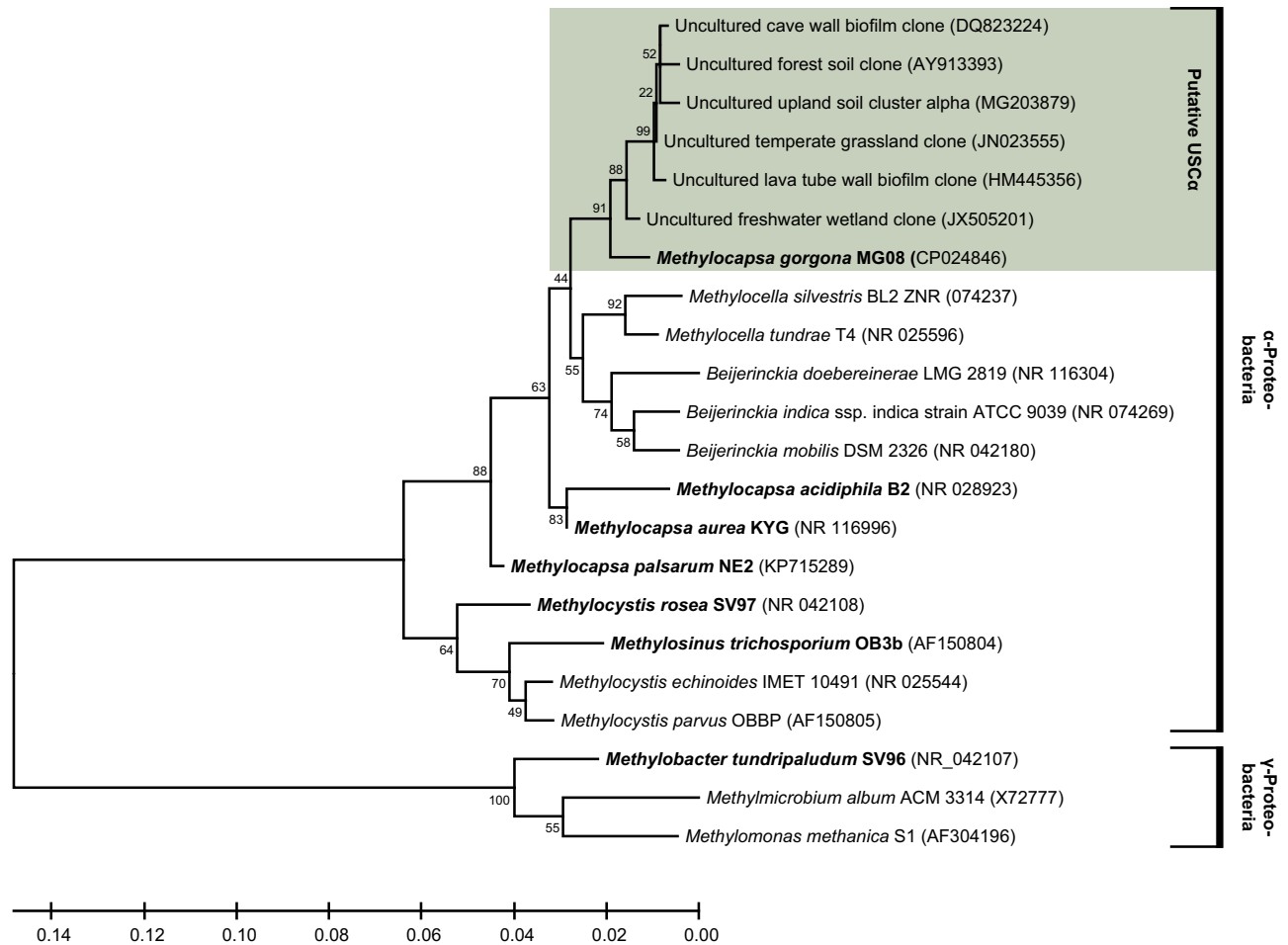

**Fig. 2 | Phylogeny of atmMOB based on 16S rRNA genes.** Unrooted maximum-likelihood tree[84,85]. Full length 16S rRNA gene based phylogenetic relationship of the respective methanotrophs and the USCα. The species investigated in this study are indicated in bold print. The percentage of trees in which the associated taxa clustered together is shown next to the branches. NCBI accession numbers are given in brackets. Putative members of the USCα are highlighted in green. Source data are provided in the Source Data file.

storage compounds. A study showing that both strains could not stay active at low $CH_4$ concentrations[26] supports our observations, but further studies are needed to clarify whether the initial colony formation is based on storage compounds.

Next, we asked whether atmMOB can obtain enough energy from growth on air to meet the basic maintenance energy of 2.8 kJ C-mol⁻¹ h⁻¹ at 20 °C postulated previously[28]. To calculate the strain-specific energy yields in C-mol, we first estimated the cellular energy yields based on the measured $CH_4$, $H_2$, and CO oxidation rates. *M. aurea* KYG yielded $6.89 \times 10^{-12}$ J cell⁻¹ h⁻¹ ($n = 5$, SD = $3.22 \times 10^{-12}$), *M. gorgona* MG08 $2.21 \times 10^{-12}$ J cell⁻¹ h⁻¹ ($n = 5$, SD = $4.36 \times 10^{-13}$), *M. palsarum* NE2 $3.09 \times 10^{-12}$ J cell⁻¹ h⁻¹ ($n = 5$, SD = $2.90 \times 10^{-13}$), and *M. rosea* SV97 $3.879 \times 10^{-12}$ J cell⁻¹ h⁻¹ ($n = 5$, SD = $1.57 \times 10^{-12}$) by the oxidation of either two or three trace gases in air (Fig. 1B). Due to the higher free energy potential and atmospheric concentration of $CH_4$ compared to atmospheric $H_2$ and CO, the energy estimates predict $CH_4$ as the major energy source for all four strains. Our estimates derive from the Gibbs free energy change of the following reactions at atmospheric conditions: $CH_4 + 2O_2 \rightarrow 2H_2O + CO_2$, $2H_2 + O_2 \rightarrow 2H_2O$, and $2CO + O_2 \rightarrow 2CO_2$ that amount to −797.4 kJ mol⁻¹, −236.8 kJ mol⁻¹, and −199.9 kJ mol⁻¹, respectively. However, these numbers do not account for the energy required for activation of $CH_4$ by the pMMO or production cost of the enzymes involved in energy conservation from the gases. While the oxidation of $H_2$ and CO is catalyzed by one enzyme[41], the oxidation of $CH_4$ to $CO_2$ by atmMOB involves at least seven

enzymes (see section: Comparative proteomics: Trace gas oxidation). Thus, due to the larger investments required for energy conservation from $CH_4$, the gases CO and $H_2$ might play more important roles as energy sources for growth on trace gases in air than indicated by the energy calculations alone. Considering the cellular dry masses and carbon content (Supplementary data file, Dataset 3) of *M. gorgona* MG08, *M. aurea* KYG, *M. palsarum* NE2, and *M. rosea* SV97, the energy yields per cell and hour translates to 0.40 kJ C-mol⁻¹ h⁻¹ (SD = 0.08 kJ C-mol⁻¹ h⁻¹), 0.71 kJ C-mol⁻¹ h⁻¹ (SD = 0.33 kJ C-mol⁻¹ h⁻¹), 0.38 kJ C-mol⁻¹ h⁻¹ (SD = 0.04 kJ C-mol⁻¹ h⁻¹), and 0.65 kJ C-mol⁻¹ h⁻¹ (SD = 0.26 kJ C-mol⁻¹ h⁻¹), at 20 °C, respectively (Fig. 3). These estimated energy yields for *M. gorgona* MG08, *M. aurea* KYG, *M. palsarum* NE2, and *M. rosea* SV97 are 3.9 – 7.4 times lower than the reported average energy requirements for cellular maintenance in aerobic bacteria (2.8 kJ C-mol⁻¹ h⁻¹ at 20 °C correspond to 4.5 kJ C-mol⁻¹ h⁻¹ at 25 °C)[29,32].

The energy estimation assumes that all cells of the filter cultures actively contributed to the observed oxidation rate. Thus, in case of inactive cells, the cellular oxidation rates and the energy yields might have been underestimated. However, while the applied cell quantification (see Methods section) considers only intact cells without assessing cellular activity, Nanoscale secondary ion mass spectrometry (NanoSIMS) based ¹⁵N incorporation confirmed activity of all measured cells (see section below: Growth on nitrogen from air), supporting our approach for energy yield estimations. However, we acknowledge that differences in activity between individual cells and potential

inaccuracies in the quantification of cells contributing to observed activities might have introduced a minor error to our estimations.

Thus, our observations contradict the basic energy premise for atmospheric $CH_4$ oxidizing bacteria[28]. Our energy estimations correspond to the similarly low energy estimates previously reported for *M. gorgona* MG08[32] and that of acetogenic and methanogenic microorganisms (0.2 kJ C·mol$^{-1}$ h$^{-1}$ at 37 °C)[42].

## Comparative proteomics

We further investigated the cellular adjustments required for life on air by *M. gorgona* MG08, *M. rosea* SV97, and *M. palsarum* NE2 by comparing the proteomes of the three strains when exposed to air (-1.9 p.p.m.v. $CH_4$) and when exposed to high $CH_4$ concentrations (-1000 p.p.m.v. $CH_4$) in air. Correspondence analyses (CA) of relative protein abundances at the two $CH_4$ concentrations revealed a clear difference in proteome allocation. This is shown by the separation of samples from the two conditions along the first CA dimension, which accounted for 79.3%, 76.5%, and 69.6% of the total inertia for *M. gorgona* MG08, *palsarum* NE2, and *M. rosea* SV97, respectively (Supplementary Fig. 3). Thus, the CA indicate that the largest shifts in the proteomes occurred as responses to changes in $CH_4$ concentration. To identify which proteins contributed most to these shifts, we identified the proteins (top 10%) with the largest contribution to the first-dimension inertias of the CA and plotted their abundances across the two conditions (Supplementary Fig. 4). We found major protein expression shifts within a variety of functional categories (based on hierarchical EggNOG[43] annotations), but for all three strains, a large proportion was related to core metabolisms, including transcription, translation, growth, energy metabolism, and amino acid and carbohydrate transport. A prominent trend observed for all three strains was that proteins associated with the categories Cell cycle control, cell division, chromosome partitioning and Cell wall/membrane/envelope biogenesis shifted towards lower relative abundances, hereafter referred to as downregulation, at atmospheric $CH_4$ compared to the 1000 p.p.m.v. $CH_4$ treatment. This pattern suggests that atmMOB lower the allocation of resources for growth when $CH_4$ availability is low, which is in line with previously reported differences in colony growth over time when comparing incubations on atmospheric and 1000 p.p.m.v. $CH_4$ concentrations[33]. We also observed major shifts in protein abundances for the categories Energy production and conversion and Carbohydrate transport and metabolism, including proteins involved in trace gas oxidation and carbon assimilation (Supplementary data file, Dataset 12–14). Based on that and the differences in trace gas oxidation patterns between the strains (Fig. 1), the relative abundances of proteins involved in trace gas oxidation, carbon assimilation via the serine cycle (Fig. 4), and the electron transport chain (Supplementary Fig. 5) were further investigated.

## Comparative proteomics: trace gas oxidation

All three strains expressed a particulate methane monooxygenase (pMMO) that catalyzes the hydroxylation of $CH_4$ to methanol ($CH_3OH$). *M. gorgona* MG08 and *M. rosea* SV97 contained higher relative abundances, hereafter referred to as upregulation, of pMMO at 1000 p.p.m.v. $CH_4$, whereas *M. palsarum* NE2 upregulated pMMO at atmospheric $CH_4$ concentrations. Furthermore, both *M. gorgona* MG08 and *M. rosea* SV97 upregulated a high affinity [NiFe] hydrogenase (Hhy), and *M. gorgona* MG08 a molybdenum-dependent carbon monoxide dehydrogenase (CODH), when exposed to air. These differences in enzyme expression patterns demonstrate different metabolic strategies to grow on air: *M. rosea* SV97 and *M. gorgona* MG08 compensate for energy limitation by upregulating enzymes for energy conservation from $H_2$ or $H_2$ and CO, while *M. palsarum* NE2 compensates for the limitation by upregulating pMMO. The Hhy increase is similar to the Hhy increase in *Mycobacterium smegmatis* enabling long-term persistence after carbon limitation[44]. The observed

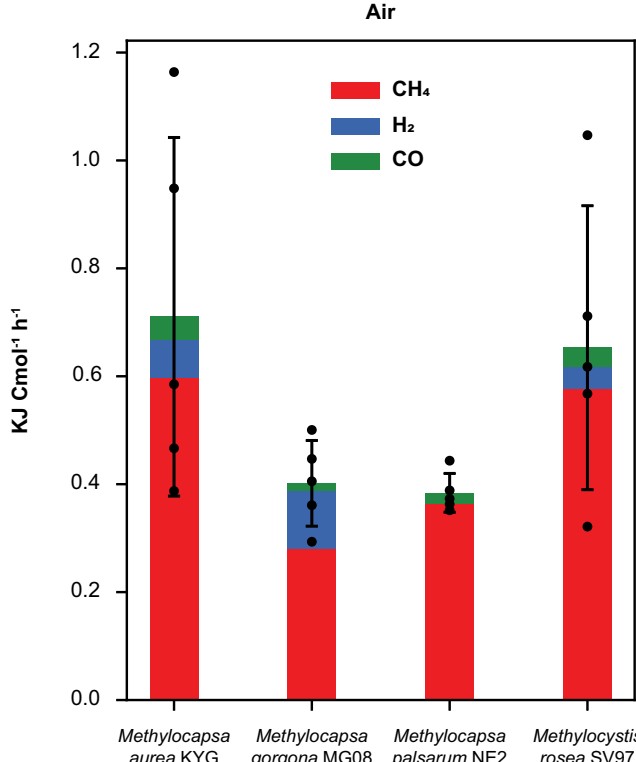

**Air**

**Fig. 3 | Energy yield from oxidation of trace gases in air.** Mean total energy yield of *M. aurea* KYG, *M. gorgona* MG08, *M. palsarum* NE2, *M. rosea* SV97 from the oxidation of atmospheric $CH_4$, $H_2$, and CO displayed in red, blue, and green, respectively, in kJ per mol cellular carbon (C-mol) and hour. Colors indicate the contribution of the individual trace gases to the total energy yield. Error bars represent the standard deviation. Black dots indicate the total energy yield from $H_2$, CO, and $CH_4$ of the respective biological replicates ($n = 5$). Source data are provided in the Source Data file and in the Supplementary Data file (Dataset 3).

upregulation of pMMO, Hhy, or CODH implies that the strains allocate resources to increase their $a_A^0$ (specific apparent affinity) for the respective trace gases as adaptations to growth on air. Additionally, the distinct strategies to grow on air by the closely related strains *M. gorgona* MG08 and *M. palsarum* NE2 suggest niche differentiation between atmMOB in nature. Furthermore, our observations of the consumption of multiple trace gases by atmMOB and adjusted resource allocation for trace gas uptake driven by changes in $CH_4$ concentration, are in line with earlier observations of a negative correlation between soil $H_2$ concentrations and the uptake of atmospheric $CH_4$[45]. Despite the ability of *M. rosea* SV97 and *M. palsarum* NE2 to oxidize CO (Fig. 1A), we were not able to determine the responsible protein complex(es) or the corresponding gene expression responses. In *M. rosea* SV97, candidate genes for CODH were identified by blasting the genome sequences against a non-redundant CODH sequence database created by using the Identical Protein Groups resource[46] (Supplementary data file, Dataset 11). However, not all the potential CODH subunits necessary to form a functional CODH were detected in the proteomes. The same blast-based approach did not result in potential candidate genes that could encode a functioning CODH in *M. palsarum* NE2 (Supplementary data file, Dataset 10). Thus, our results indicate that distantly related or previously undiscovered enzymes catalyze atmospheric CO oxidation in *M. palsarum* NE2.

The expression of the methanol dehydrogenase (MDH), which catalyzes the second step in $CH_4$ oxidation, the oxidation of methanol ($CH_3OH$) to formaldehyde ($CH_2O$), matched the pMMO expression patterns for all three strains. This indicates a close interaction between these two enzymes as previously suggested for *Methylococcus*

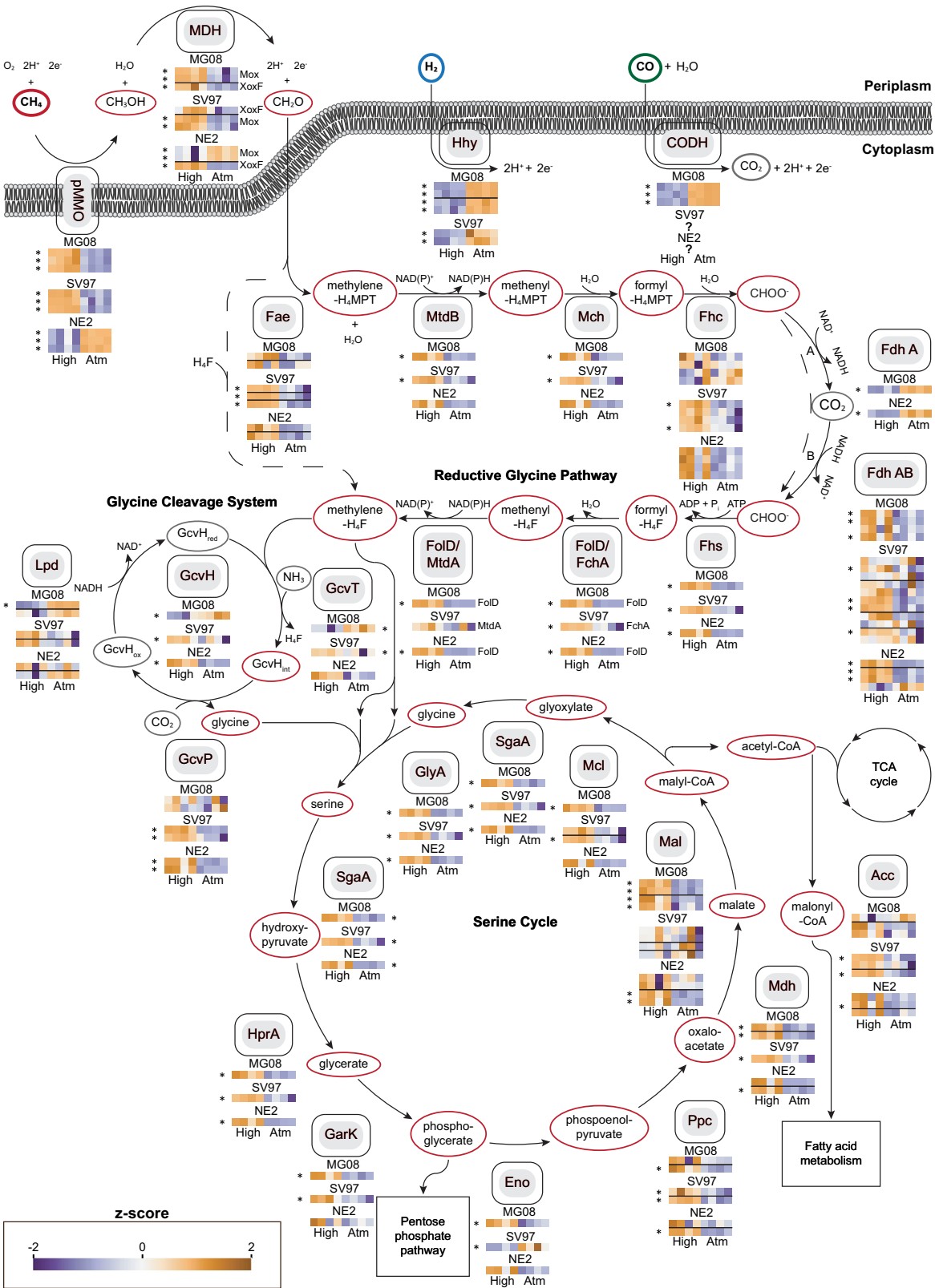

*capsulatus*[47]. MDHs were upregulated at high $CH_4$ concentrations by *M. gorgona* MG08 and *M. rosea* SV97, but not by *M. palsarum* NE2 (Fig. 4). Only the putative lanthanide-dependent methanol dehydrogenase (XoxF) of *M. palsarum* NE2 did not follow the pMMO pattern.

Formaldehyde is a key intermediate of both catabolism and anabolism in methanotrophs. The enzymes involved in the catabolic oxidation of formaldehyde via methylene-tetrahydromethanopterin

(-$H_4MPT$), methenyl-$H_4MPT$, and formyl-$H_4MPT$ to formate (CHOO⁻), were downregulated in the air treatment of the three strains (Fig. 4). The toxic and highly reactive formaldehyde condenses spontaneously to methylene-$H_4MPT$[48] and methylenetetrahydrofolate ($H_4F$)[49] and thus, higher concentrations of formaldehyde activating enzymes (Fae) may not be required for the oxidation of formaldehyde during growth on air. This may explain why enzymes involved in formaldehyde

**Fig. 4 | Metabolic adjustments during growth on air.** Comparative proteomics of *M. gorgona* MG08, *M. rosea* SV97, and *M. palsarum* NE2 exposed to 1000 p.p.m.v. $CH_4$ (High) in air and 1.9 p.p.m.v. (Atm) $CH_4$ in air. Normalized and standardized expression of enzymes involved in the central carbon and energy metabolism is shown. High relative abundance = orange, low relative abundance expression = blue. * indicates significant difference in expression between treatments (two-sided t-test). Horizontal lines in the heatmaps separate operons of enzymes catalyzing the same reaction. ? = unknown enzyme. Abbreviations: pMMO—particulate methane monooxygenase, MDH—methanol dehydrogenase (Mox) and (XoxF), Hhy– [NiFe] hydrogenase, CODH—carbon monoxide dehydrogenase, Fae—formaldehyde activating enzyme, MtdB—NAD(P)-dependent methylene-tetrahydromethanopterin dehydrogenase, Mch—methenyl-tetrahydromethanopterin cyclohydrolase, Fhc—formyltransferase/hydrolase, Fdh A—NAD-dependent formate-dehydrogenase, Fdh AB—molybdopterin dependent formate dehydrogenase, Fhs—formate-tetrahydrofolate ligase, FolD—bifunctional methylene-tetrahydrofolate dehydrogenase/methenyl-tetrahydrofolate cyclohydrolase, FchA—methenyl-tetrahydrofolate cyclohydrolase, MtdA—NADP-dependent methylene-tetrahydrofolate dehydrogenase, GcvH—glycine cleavage system H protein, GcvT—aminomethyltransferase, GcvP—glycine dehydrogenase, Lpd—dihydrolipoyl dehydrogenase, GlyA—serine hydroxymethyltransferase, SgaA—serine-glyoxylate aminotransferase, HprA—glycerate dehydrogenase, GarK - 2-glycerate kinase, Eno—enolase, Ppc—phosphoenolpyruvate carboxylase, Mdh—malate dehydrogenase, Mal—malate-CoA ligase, Mcl—L-malyl-CoA lyase, Acc—Acetyl-CoA carboxylase, $NAD^+$—nicotinamide adenine dinucleotide, $NADP^+$—nicotinamide adenine dinucleotide phosphate, ATP—adenosine triphosphate. Source data are provided in the Source Data file and in the Supplementary Data file (Dataset 5).

oxidation, despite the upregulation of pMMO and MDH, were downregulated in *M. palsarum* NE2 at atmospheric $CH_4$ concentrations. At high $CH_4$ concentrations, the upregulation of Fae and downstream enzymes for further oxidation to formate could represent a cellular detoxification mechanism to avoid high cellular formaldehyde concentrations and to increase NADH synthesis[50].

Two different formate dehydrogenases (Fdh A and Fdh AB) that catalyze the energy-conserving oxidation of formate to $CO_2$ were expressed by *M. gorgona* MG08 and M. *palsarum* NE2 (Fig. 4). The metal-free and potentially irreversible $NAD^+$-dependent Fdh A[33,51] was upregulated at atmospheric $CH_4$ concentrations in *M. gorgona* MG08 and M. *palsarum* NE2. Possibly, the upregulation of a one-directional enzyme minimizes back-flow of $CO_2$ to formate at high $CO_2$ concentrations and thus prevents energy loss by $CO_2$ reduction. Fdh AB, which was expressed by all three strains, is molybdopterin-dependent and homologous to the reversible Fdh found in *Rhodobacter capsulatus*[52]. It catalyzes, in addition to formate oxidation, the reduction of $CO_2$ to formate[52,53]. The Fdh AB was upregulated in all three strains at high $CH_4$ concentration. Under these conditions, when excess reducing power is available, the Fdh AB might enable the reduction of $CO_2$ for carbon assimilation via the reductive glycine pathway and serine cycle, but further investigations are needed to test whether $CO_2$ reduction to formate truly occurs in atmMOB.

## Comparative proteomics: carbon assimilation

Carbon assimilation was downregulated in all three strains at atmospheric $CH_4$ concentrations. In the three strains, formaldehyde can condense with $H_4F$ to methylene-$H_4F$ and then be further assimilated through the glycine cleavage system and serine pathway. Additionally, formaldehyde can be first oxidized via the $H_4MPT$-mediated pathway to formate and then enter, instead of being oxidize to $CO_2$, the $H_4F$-mediated reductive pathway to methylene-$H_4F$ (Fig. 4). The increased expression of enzymes involved in the reduction of formate via formyl-$H_4F$ and methenyl-$H_4F$ to methylene-$H_4F$ in the high $CH_4$ treatment indicates an enhanced investment into carbon assimilation. Since formaldehyde spontaneously condenses with $H_4F$ to methylene-$H_4F$, the upregulated enzymes for $H_4F$-mediated formaldehyde oxidation could also contribute to formate formation. However, experiments with *Methylorubrum extorquens* demonstrated that formaldehyde oxidation occurred through its $H_4MPT$-mediated pathway while the reductive pathway via formate to methylene-$H_4F$ represented the major assimilatory flux[50,54]. Therefore, we consider this as the most likely explanation for the expression patterns in *M. gorgona* MG08, *M. rosea* SV97, and *M. palsarum* NE2.

The glycine cleavage/synthase system (GCS)[55,56] catalyzes the oxidative cleavage of glycine to NADH, $NH_3$, $CO_2$, and a methylene group. It also catalyzes the reverse reaction, the synthesis of glycine from NADH, $NH_3$, $CO_2$, and methylene-$H_4F$. The upregulation of the GCS by *M. rosea* SV97 and *M. palsarum* NE2 at high $CH_4$ concentrations suggests an increased synthase activity and thus increased carbon assimilation, corresponding to the overall upregulation of the reductive glycine pathway (Fig. 4). However, we lack a plausible explanation for the increased expression of the GCS by *M. gorgona* MG08 at atmospheric $CH_4$ concentrations.

As the next step in carbon assimilation, the bidirectional serine hydroxymethyltransferase (GlyA) condenses glycine with methylene-$H_4F$ to serine, representing the first step of the serine cycle (Fig. 4). The enzymes involved in the serine cycle were upregulated by the strains when exposed to high $CH_4$ concentrations, indicating investment into carbon assimilation at this condition.

## Comparative proteomics: electron transport chain

The relative abundances of protein complexes involved in the electron transport chain for ATP synthesis was lower in atmospheric $CH_4$ compared to the high $CH_4$ treatment (Supplementary Fig. 5). The NADH-quinone oxidoreductase, cytochrome c oxidase, and the ATP synthase were all highly expressed at high $CH_4$ concentrations indicating increased investment into energy conservation at high substrate supply. We propose that to overcome energy limitations when exposed to air, all three strains upregulate the expression of enzymes for the oxidation of at least one trace gas to maximize uptake rates and energy yield, while investments in energy conservation and energy-intensive carbon assimilation are reduced. This reduced investment in assimilation is in line with the low concentrations of the trace gases, low uptake rates, and overall slow growth of atmMOB when incubated with air as energy and carbon source[33].

## Specific affinity

The $a_A^0$ for $CH_4$, expressed as the fraction of $V_{max(app)}$ and $K_{m(app)}$, directly represents the capacity to oxidize $CH_4$ at low concentrations. To test if the pMMO upregulation in *M. palsarum* NE2 and downregulation in *M. gorgona* MG08 at atmospheric $CH_4$ concentrations translate into a higher $a_A^0$ for $CH_4$ by *M. palsarum* NE2 compared to *M gorgona* MG08, we measured the $a_A^0$ of the two strains when grown on air. To avoid increases in the apparent half saturation constant ($K_{m(app)}$) estimates at high $CH_4$ concentrations[57], we pre-incubated the cultures on filters floating on carbon-free medium for five months with air as sole energy and carbon source.

By measuring the $CH_4$ oxidation per cell and hour at different $CH_4$ concentrations, we estimated a $K_{m(app)}$ of 48.54 nM $CH_4$ and a $V_{max(app)}$ of $4.91 \times 10^{-8}$ nmol $cell^{-1}$ $h^{-1}$ resulting in a $a_A^0$ of $1.01 \times 10^{-9}$ L $cell^{-1}$ $h^{-1}$ for *M. gorgona* MG08 (Fig. 5). A $K_{m(app)}$ of 48.54 nM $CH_4$ is within the range of the $K_{m(app)}$ values measured for fresh oxic soils reported by Bender and Conrad (30 – 51 nM)[13], which led to the theory that atmMOB are oligotrophs with a high affinity for $CH_4$. This is the first observation of a methanotroph in pure culture showing a $K_{m(app)}$ that is in the range of the low $K_{m(app)}$ values measured in upland soils[25,58]. Our $a_A^0$ estimate for *M. gorgona* MG08 ($1.01 \times 10^{-9}$ L $cell^{-1}$ $h^{-1}$) is approximately five times higher than the $a_A^0$

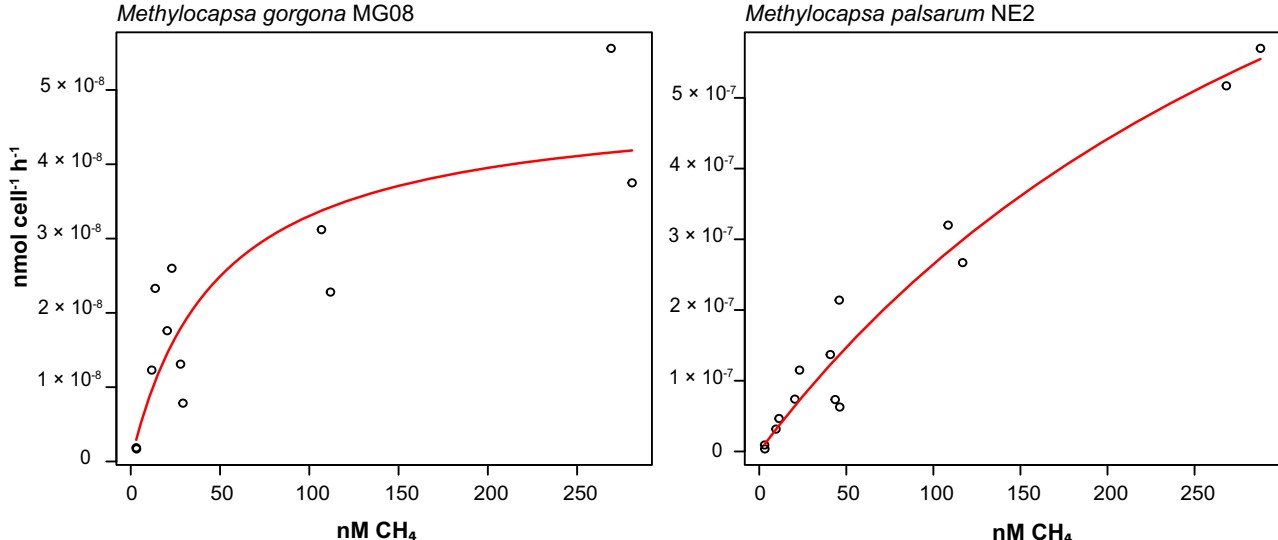

**Fig. 5 | CH₄ oxidation kinetics.** Michaelis-Menten hyperbolic curve (displayed in red) fitted to the oxidation rate per cell of *M. gorgona* MG08 and *M. palsarum* NE2 at CH₄ concentrations ranging between 3 nM and 287 nM. The nM CH₄ display CH₄ concentrations dissolved in water and correlate with the partial pressure of CH₄ above the water. At 20 °C and atmospheric pressure (1.013 bar), 2.98, 50, 100, and 250 nM CH₄ dissolved in water correspond to 1.9, 31.9, 63.7, and 159.3 p.p.m.v. in air. Source data are provided in the Source Data file and in the Supplementary Data file (Dataset 9).

reported by Tveit et al. $1.95 \times 10^{-10}$ L cell$^{-1}$ h$^{-1}$ [33]. However, the $K_{m(app)}$ and $V_{max(app)}$ reported by Tveit et al. amount 4905 nM and $95.4 \times 10^{-8}$ nmol cell$^{-1}$ h$^{-1}$, respectively, values approximately 100 and 20 times higher than in the current study. These differences might derive from the use of liquid cultures pre-incubated at 20% CH₄ by Tveit et al. Such a high CH₄ concentration could have influenced the cellular pMMO concentration, as indicated by the pMMO upregulation by *M. gorgona* MG08 at 1000 p.p.m.v. CH₄ described above (Fig. 4), and thus increased the $K_{m(app)}$ and $V_{max(app)}$ estimates. Dunfield and Conrad reported a similar alteration of $K_{m(app)}$ and $V_{max(app)}$ for *Methylocystis* strain LR1 after comparing starved cells to cells exposed to 10% CH₄ while the $a_A^0$ was more constant[57].

The $K_{m(app)}$ of *M. palsarum* NE2 was 402.08 nM CH₄, which is approximately eight times higher than the $K_{m(app)}$ of *M. gorgona* MG08. Despite this higher $K_{m(app)}$, the $a_A^0$ of *M. palsarum* NE2 was $3.30 \times 10^{-9}$ L cell$^{-1}$ h$^{-1}$, three times higher than estimated for *M. gorgona* MG08. This high $a_A^0$ derives from its substantially higher $V_{max(app)}$ of $133 \times 10^{-8}$ nmol cell$^{-1}$ h$^{-1}$ (Fig. 5) and aligns with the proteomic data. The upregulation of the pMMO at atmospheric CH₄ concentrations by *M. palsarum* NE2 seem to translate into higher a $a_A^0$ for CH₄ compared to *M. gorgona* MG08 that downregulated its pMMO. This is also reflected in the CH₄ oxidation rate of *M. palsarum* NE2 at atmospheric concentrations, which surpassed the rate of *M. gorgona* MG08, despite having a lower apparent affinity for CH₄ (Fig. 5).

An apparent affinity of 402.08 nM CH₄ is not considered as high affinity[28,41]. Thus, as both strains grow on air, a high apparent affinity for CH₄ cannot be considered a prerequisite for this lifestyle. The $a_A^0$ of *M. gorgona* MG08 and *M. palsarum* NE2 are approximately equally high and three times higher, respectively, than the recently reported $a_A^0$ of *Methylotuvimicrobium buryatense* 5GB1C[59] and 30 and 100 times higher, respectively, than the $a_A^0$ of *Methylocystis* sp. SC2 ($3.4 \times 10^{-11}$ L cell$^{-1}$ h$^{-1}$)[26], the MOB with the fourth highest $a_A^0$ for CH₄ known so far. However, the different experimental setups and CH₄ concentrations used to determine $a_A^0$ might render the comparisons invalid[26,59]. Nevertheless, our results show that the specific affinity ($a_A^0$), rather than the affinity ($K_{m(app)}$), is the appropriate model to determine the efficiency of atmospheric CH₄ utilization by atmMOB. This is in line with the work on oligotrophic substrate uptake at low concentrations by Button[60].

## Growth on nitrogen from air

The four atmMOB, *M. gorgona* MG08, *M. palsarum* NE2, *M. rosea* SV97, and *M. aurea* KYG, encode all genes required for dinitrogen (N₂) fixation[33] and grow in nitrogen-free medium at high CH₄ concentrations[33,36,37,38]. To test the potential for growth on nitrogen from air, we incubated filter cultures of these strains with air as the sole energy, carbon, and nitrogen source. The colony formation of all four strains after three months, and trace gas uptake by *M. gorgona* MG08 after one year, demonstrates growth in the absence of bioavailable nitrogen sources in the medium (Fig. 6A, B, and Supplementary Fig. 6). This suggests that all four strains can either cover their nitrogen requirements by the fixation of N₂, by the utilization of atmospheric reactive nitrogen during growth on air, or both. While the N₂ concentration in air is approximately 78 %, ammonia concentrations, for example, have been observed to range between 0.2 and 24 p.p.b.v[61].

To test for N₂ fixation during growth on air by *M. gorgona* MG08, we incubated filter cultures in air enriched with ¹⁵N-N₂ (~23 atom % (at %) of the total N₂) and CH₄, H₂, and CO concentrations fluctuating between 0.03 and 3 p.p.m.v. for two months. Afterwards, we measured cellular ¹⁵N₂ fixation using NanoSIMS (Fig. 6C). All cells (n = 379) measured during NanoSIMS incorporated ¹⁵N (Supplementary Dataset 15 and 16) indicating that all cells have been active during incubation. This supports the validity of our energy estimations (as mentioned above) and demonstrates growth on trace gases in air. The cellular ¹⁵N ranged from 2.99 at% to 61.89 at% with an average of 30.79 at% (SD = 7.82 at%) (Supplementary data file, Dataset 15) while the control without ¹⁵N₂ enrichment averaged at 0.37 at% (SD = 0.04) (Supplementary data file, Dataset 16). Since the ¹⁵N₂ in the headspace amounted to approximately 23 at% of the total N₂ during incubation, the ¹⁵N average of the cells should not have amounted to more than 23 at%. The high values might issue from bioavailable ¹⁵N-species in the ¹⁵N₂ gas used for incubation[62]. Purity tests of the ¹⁵N₂ gas prior to the ¹⁵N₂ fixation experiments revealed only a minor NOₓ contamination of 101.59 (nmol ml$^{-1}$) with a very low ¹⁵N fraction of 0.0014 at% and ammonia levels below detection limit (1 nmol ml$^{-1}$), and thus do not provide indications that contamination can explain our observations. However, even undetectable trace amounts of ¹⁵N ammonia contaminating the ¹⁵N₂ might be sufficient to cause high

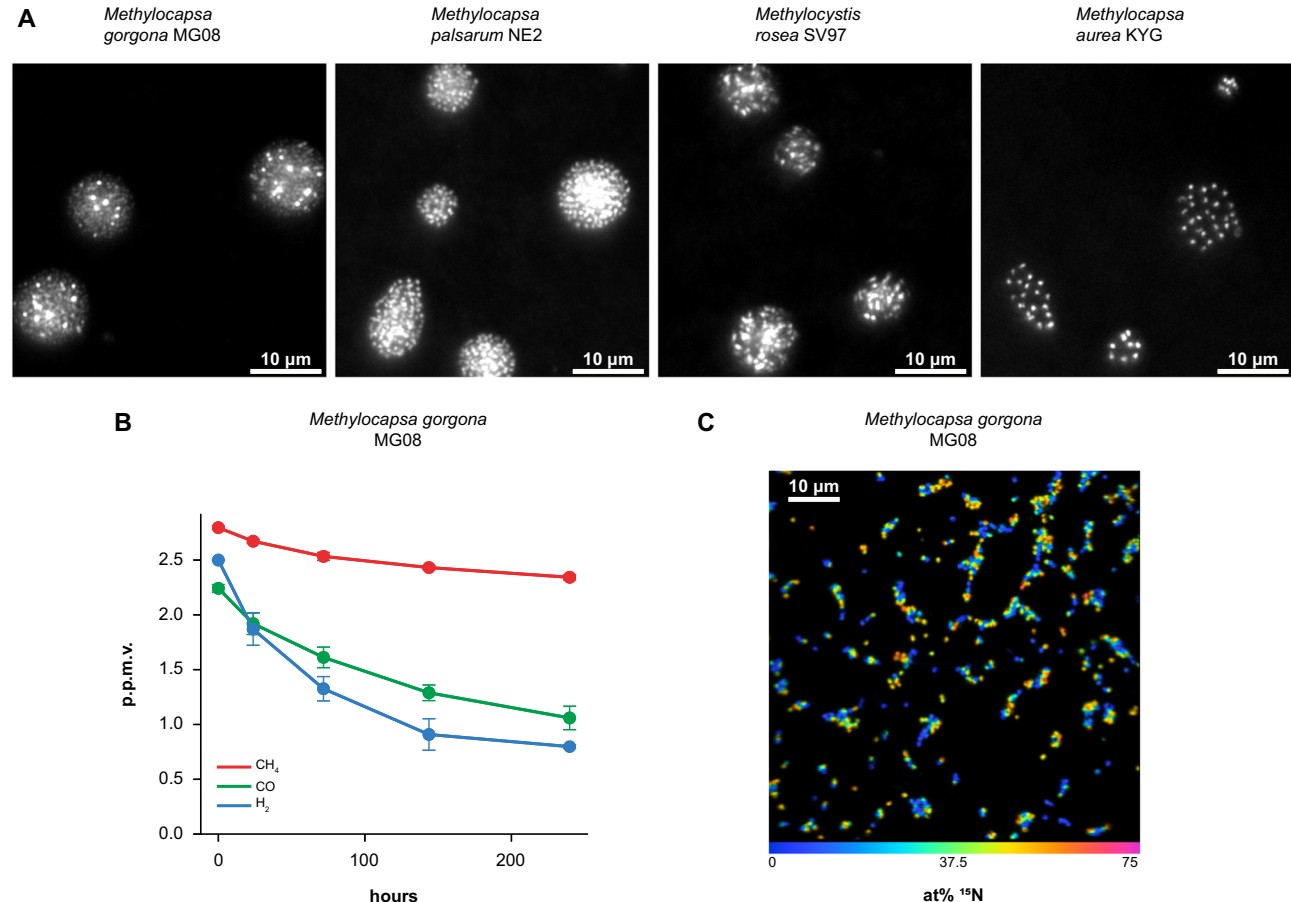

**Fig. 6 | Growth with air as nitrogen source. A** Images of SYBR green I stained colonies formed by *M. gorgona* MG08, *M. palsarum* NE2, *M. rosea* SV97, and *M. aurea* KYG after three months of incubation with air as sole energy, carbon, and nitrogen source. Colony formation experiments with air as sole energy, carbon, and nitrogen source have been repeated independently for at least two times per strain all leading to similar results. **B** Trace gas oxidation with air as nitrogen source. Mean trace gas oxidation of *M. gorgona* MG08 filter cultures after 12 months of incubation with air as sole carbon, energy, and nitrogen source. Error bars indicate standard deviation between the biological replicates ($n = 4$). Source data are provided in the Source Data file and in the Supplementary Data file (Dataset 1). **C** $^{15}$N incorporation on air. NanoSIMS visualization of $^{15}$N incorporation by *M. gorgona* MG08 after two months of incubation under a $^{15}$N$_2$-enriched atmosphere. Fraction values of $^{15}$N/($^{14}$N + $^{15}$N) are given in at%. Different colors represent the cellular $^{15}$N content as displayed by the color bar. The $^{15}$N incorporation on air experiment has not been repeated. Data are provided in the Supplementary Data file (Dataset 15).

cellular $^{15}$N-enrichments given the low amount of biomass of the filter cultures and the long incubation times. Thus, despite not being able to detect any $^{15}$N-contaminants, we cannot exclude contamination and thereby cannot confirm N$_2$ fixation. However, considering the growth on nitrogen-free medium (Supplementary Fig. 6) and the NanoSIMS experiment, we can conclude that *M. gorgona* MG08 either fixes N$_2$ or trace concentrations of reactive nitrogen species. This demonstrates that atmMOB can cover their nitrogen requirements for growth, in addition to energy and carbon, from air. Additionally, it suggests that atmMOB may not be nitrogen limited under most natural conditions, partially answering the question by Bodelier and Steenberg regarding conditions that can be nitrogen limiting to atmMOB[39].

## Implications of the findings

Overall, the results demonstrate that growth on air and enduring oxidation of atmospheric CH$_4$ is not restricted to members of the clades USCα and γ, but more widespread than previously assumed. Our data show that former liquid culture-based cultivation approaches lead to an underestimation of the true potential of methanotrophic pure cultures to oxidize atmospheric CH$_4$. The appearance of atmMOB outside the USCα and γ revises our understanding of the biological atmospheric CH$_4$ sink and should be considered in future studies. The strain specific oxidation patterns and the estimations of the $a_A^0$ for CH$_4$ demonstrate that both the mixotrophic oxidation of atmospheric trace gases and a high $a_A^0$ for CH$_4$ are key to obtain sufficient energy for growth on air. The estimated energy requirements for growth of the four atmMOB are substantially lower than the maintenance energy value used as basic premise for an oligotrophic lifestyle of MOB. Additionally, atmMOB seem to cover not only their energy and carbon but also their nitrogen requirements from the atmosphere. This opens a new perspective on physiological limitations of atmospheric trace gas oxidizers and suggests that atmMOB may carry an ideal set of properties needed for pioneering species to initiate primary succession in unfavorable environments. Additionally, the high $a_A^0$ for CH$_4$ enables atmMOB to utilize trace concentrations of CH$_4$ as energy and carbon source for growth while being extremely oligotrophic. This bears the potential for cost-effective and efficient biofiltration of anthropogenic emissions containing CH$_4$ concentrations far below the lower explosive limit. The common metabolic strategy to grow on air seems to be the downregulation of enzymes involved in energy-intensive processes combined with the upregulation of enzymes for the oxidation of trace gases. However, the differing expression patterns of enzymes for trace gas oxidation and the strain-specific trace gas oxidation patterns indicate that a diverse metabolic repertoire has evolved to enable life on air.

## Methods

### Filter-based cultivation with air as the sole carbon and energy source

The strains were pre-incubated in liquid cultures in 100 mL serum bottles containing 10 mL of 10x diluted, EDTA-free NMS medium (DSMZ medium 921 with 10x the iron concentration and 1 μM lanthanum) and a headspace of 20% $CH_4$ in air. The Fe-EDTA stated in the original DSMZ recipe was substituted with $FeSO_4$. Depending on the strain, the medium was adjusted to a pH of 6.8 (*Methylocapsa gorgona* MG08, *Methylosinus trichosporium* OB3b, *Methylocystis rosea* SV97, *Methylobacter tundripaludum* SV96) or 5.8 (*Methylocapsa aurea* KYG, *Methylocapsa acidiphila* B2, and *Methylocapsa palsarum* NE2). The serum bottles were sealed using butyl rubber stoppers (Chromacol 20-B3P, Thermo Fisher Scientific, Waltham, Massachusetts, USA) and crimp caps, and incubated at 20 °C in dark, until the strains were in an exponential growth phase. Strain purity was controlled routinely via microscopy and by confirming the absence of heterotrophic growth on agar plates with a rich medium containing tryptone, yeast extract, and glucose[32]. After reaching exponential growth phase, the strains were diluted with the described medium and filtered on 25 mm polycarbonate (PC) filters (Whatman 10417006, Cytiva, Massachusetts, Marlborough, USA) using a filtration manifold (EQU-FM-10X20-SET, DHI, Hørsholm, Denmark). The final cell density on the filters amounted approximately 20 cells per photo area (630x magnification). The filters were transferred into Petri dishes containing 10x diluted, EDTA-free NMS medium (DSMZ medium 921 with 10x the iron concentration and 1 μM lanthanum) and left floating on the medium with the cells facing upwards. The Petri dishes enable constant ventilation of the filter cultures with ambient air while keeping the cultures sterile. The cultures were incubated at 20 °C in dark, with air as the sole energy and carbon source, for at least three months until initiation of experiments. All cultivation steps were carried out under sterile conditions. Before experiments, colony formation and microcolony morphology of the strains was routinely checked via microscopy as previously described[33]. To verify that atmMOB grow on the three trace gases $CH_4$, CO, and $H_2$ in air, we incubated *M. gorgona* MG08 cells on filters floating on 10x diluted EDTA-free NMS medium in glass bottles sealed with Safety-Caps (JR-S-11011, VICI AG International, Schenkon, Switzerland) under two different atmospheres: One of synthetic air without the trace gases $CH_4$, CO, and $H_2$ (Gas composition: 400 p.p.m.v. $CO_2$ and 20.9% $O_2$ in $N_2$) and another of ambient air. After an incubation period of 15, 20, and 50 days at 20 °C, the growth of cells in the different treatments was compared via microcopy (Supplementary Fig. 2). To create the synthetic air atmosphere, gas was added to the bottles as described in the section Trace gas oxidation experiments.

### Trace gas oxidation experiments

For all trace gas oxidation experiments, filter cultures of the respective strains, pre-incubated with air as the sole carbon and energy source, were transferred into 250 mL glass bottles containing 50 mL 10x diluted EDTA-free NMS medium (DSMZ medium 921 with 10x the iron concentration and 1 μM lanthanum). The bottles were capped with Safety-Caps (JR-S-11011, VICI AG International). To create a defined atmosphere within the bottles, the headspace of each bottle was flushed for 10 minutes with high-purity synthetic air containing 400 p.p.m.v. $CO_2$ (HiQ, AGA, Sweden) using a gassing manifold. The headspace pressure was adjusted to approximately 1.05 bar. Then, 1 mL 1000 p.p.m.v. $CH_4$ in $N_2$ (HiQ, AGA, Sweden), 1 mL 1000 p.p.m.v. $H_2$ in $N_2$ (HiQ, AGA, Sweden), and 1 mL 1000 p.p.m.v. CO in $N_2$ (HiQ, AGA, Sweden) were added to the headspace using a gas tight syringe. To assess the oxidation rate of the respective strains, the cultures were incubated at 20 °C and the change in $CH_4$, $H_2$, and CO concentrations within bottle headspaces were measured. For each measurement, 2 mL of the gas in the headspace were sampled using a gas tight syringe.

The gas samples were analyzed with a gas chromatograph (Thermo-Scientific Trace 1300, Thermo Fisher Scientific) equipped with a sample loop, a Hayesep column (SU12875, RESTEK, Bellefonte, Pennsylvania, USA), a Molsieve 5 A column (PKC17080, RESTEK), a pulsed discharge detector, and a flame ionization detector. A high-quality gas containing 2.5 p.p.m.v. $CH_4$, 2.5 p.p.m.v. $H_2$, and 2.5 p.p.m.v. CO in $N_2$ served as standard (HiQ, AGA, Sweden). To determine the gas concentrations in the bottle headspaces, standard curves were created every day of measurement. The headspace pressure was measured at the end of the experiment using a manometer (LEO1, Keller, Winterthur, Switzerland). The mass of each trace gas was calculated by applying the ideal gas law and adjusted for changes in pressure caused by gas removal during measurement.

### Cell quantification

For quantification, cells from filters were processed as follows: The filters were placed on 150 μL 25x SYBR green I (10463252, Thermo Fisher Scientific) with cells facing upwards and incubated for 10 min in dark. Next, the cells were washed twice by letting the filters float on milliQ water for 5 min in dark. To remove the cells from the filters, the filters were transferred into 5 mL tubes (0030122321, Eppendorf, Hamburg, Germany) containing 2 mL 10x diluted solution 1 (DSMZ medium 921), so that the filters sticked to the tube wall and the cells faced inwards[63]. The tubes were vortexed for 10 min at medium speed. After vortexing, remaining cells on the filters were washed off and collected by rinsing the filters with 1 mL solution 1 (DSMZ medium 921). The washed filters were dried in dark, and the cell removal controlled via fluorescence microscopy. The 3 mL cell suspension resulting from the cell removal was spiked with 50 μL absolute counting beads (Invitrogen™ CountBright™ Plus, Thermo Fisher Scientific). After, the cells in suspension were immediately counted using a flow cytometer (BD FACSAria III, BD Biosciences, Franklin Lakes, New Jersey, USA). The flow cytometer was set up to capture SYBR Green I (excitation = 498 nm, emission = 522 nm) in the green channel and the Invitrogen™ CountBright™ Plus Absolute Counting Beads (excitation = 350−810 nm, emission = 385−860 nm) in the blue channel. Unstained cells of *M. gorgona* MG08, *M. palsarum* NE2, *M. rosea* SV97, and *M. aurea* KYG were used to define the autofluorescence threshold in the different channels. Size beads (NPPS-4K, Spherotech, Lake Forest, Illinois, USA) were used to draw the forward scatter gate covering events between 0.58 and 4 μm (Supplementary Fig. 10). SYBR green I stained cells and absolute counting beads were counted to determine the cell number within filter cultures, using the following equation:

$$x_{filter} = \frac{b_{abs}}{b_c} \times x_c \tag{1}$$

Where $x_{filter}$ is the absolute cell number of the filter culture, $b_{abs}$ the number of counting beads added, $b_c$ the number of counting beads counted, and $x_c$ the number of SYBR green I stained cells counted.

### Cell-specific oxidation rates and energy calculation

The gas-specific oxidation rates of the different strains at atmospheric $CH_4$, $H_2$, and CO concentrations and pseudo first order kinetics were calculated using the ideal gas law and the integrated rate law for first order reactions (calculations can be found in the Supplementary data file (Dataset 2)). The oxidation rate per cell was calculated by dividing the oxidation rate of the filter culture by the corresponding cell number. The Gibbs free energy changes ($\Delta_r G$) for the following reactions: $CH_4 + 2O_2 \rightarrow 2H_2O + CO_2$, $2H_2 + O_2 \rightarrow 2H_2O$, $2CO + O_2 \rightarrow 2CO_2$, at 20 °C, 1.013 bar absolute, and atmospheric concentrations of $CH_4$ (1.87 p.p.m.v.), $H_2$ (0.5 p.p.m.v.), and CO (0.2 p.p.m.v.) amount to −797.4 kJ mol⁻¹, −236.8 kJ mol⁻¹, and −199.9 kJ mol⁻¹ respectively. The values are based on the values for Gibbs free energy of formation found in literature[32,64] and the

following equation:

$$\Delta_r G = \Delta_r G^\circ + RT \ln Q_r \qquad (2)$$

where $\Delta_r G^\circ$ is the Gibbs free energy change at standard conditions, $R$ the gas constant, $T$ the temperature, ln the natural logarithm, and $Q_r$ the reaction quotient. The estimates of energy yield per cell were obtained from the trace gas oxidation per cell and the $\Delta_r G$. The energy yield from trace gas oxidation per carbon mol of biomass and hour (kJ C-mol$^{-1}$ hour$^{-1}$) was estimated including the strain specific carbon content and dry weight as previously shown[32].

## Cellular dry mass and carbon content estimations

The strain-specific cellular dry mass was determined by measuring single-cell buoyant mass distributions of the strains in H$_2$O-based and deuterium oxide (D$_2$O)-based solutions of phosphate saline buffer (PBS) using a suspended microchannel resonator (SMR)[65,66]. To do so, filter cultures of *M. gorgona* MG08, *M. rosea* SV97, *M. palsarum* NE2, and *M. aurea* KYG were incubated on 10x diluted, EDTA-free NMS medium (DSMZ medium 921 with 10x the iron concentration and 1 μM lanthanum) under an atmosphere of 1000 p.p.m.v. CH$_4$ in air for three weeks. After, cells of filter cultures were fixed by incubating the filters on 150 μl formaldehyde in 1×PBS (4% w/v) for an hour at room temperature. The cells were washed twice by letting the filters float on water for 5 min. The fixed cells were harvested as described in the section called Cell quantification. The resulting cell suspension was stored at 4 °C until measurements of the buoyant cell mass. For the measurements, two aliquots with the same volume of each cell suspension were created. The water of the aliquots was replaced by H$_2$O-based (1x PBS in H$_2$O) and D$_2$O-based (1x PBS in 9:1 D$_2$O:H$_2$O) solutions of known density (1.0043 g cm$^{-3}$ and 1.1033 g cm$^{-3}$ at 20 °C, respectively). To do so the aliquots were dried in a vacuum concentrator at 37 °C. After, the cells of one of the two aliquots were resuspended in 50 μl of the H$_2$O-based solution. The cells of the second aliquot were resuspended in 50 μl of the D$_2$O-based solution. The buoyant mass of the cells in the aliquots were measured with a SMR (LifeScale, Affinity Biosensors, Santa Barbara, California, USA). The precision and accuracy of the SMR was verified by creating calibration curves using NIST-certified polystyrene beads (ThermoFisher Scientific) as performed previously[67].

The buoyant mass data were exported from the LifeScale instrument and further analyzed using the Python 3[68] version 3.9.13 and the packages Pandas[69] version 1.4.4, Matplotlib[70] version 3.5.2, and Seaborn[71] version 0.11.2 (Supplementary Fig. 9). The dry mass of the strains was calculated as described[65] using the median of the single-cell buoyant mass distributions in H$_2$O-based and D$_2$O-based solutions and the following equation:

$$m_{dry} = \frac{\rho_{D_2O} \times m_{b.H_2O} - \rho_{H_2O} \times m_{b.d_2O}}{\rho_{D_2O} - \rho_{H_2O}} \qquad (3)$$

Where $m_{dry}$ is the dry mass, $\rho_{D_2O}$ the density of the D$_2$O-based solution, $m_{b.H_2O}$ the cell's buoyant mass in the H$_2$O-based solution, $\rho_{H_2O}$ the density of the H$_2$O-based solution, and $m_{b.d_2O}$ the cell's buoyant mass in the D$_2$O-based solution.

The cellular carbon contents of the atmMOB were analyzed using an elemental analyzer coupled to an isotope ratio mass spectrometer (EA-IRMS; EA1110 coupled via a ConFlo III interface to a DeltaPLUS IRMS, Thermo Fisher Scientific). Biomass for the carbon content analysis was derived from stirred-tank bioreactor (DASbox® Mini Bioreactor System, Eppendorf) cultures. The cultures were grown in 10x diluted, EDTA-free NMS medium (DSMZ medium 921 with 10x the iron concentration and 1 μM lanthanum) at 80 rpm (marine impeller), 20 °C, pH 6.8 (CO$_2$ controlled) and a gassing rate of 0.24 vessel volumes per minute (6000 p.p.m.v. CH$_4$ in air) using a microsparger

(78530205, Eppendorf). After harvest, cultures were washed three times with milliQ water and lyophilized.

## Comparative proteomics

For the atmospheric CH$_4$ treatment, filter-cultures of *M. gorgona* MG08, *M. rosea* SV97 and *M. palsarum* NE2 were incubated with air as the only carbon and energy source for seven months. After, the cells on filters were harvested as described in the section called Cell quantification, lyophilized, and stored at −80 °C until further processing. The 1000 p.p.m.v. CH$_4$ treatment of the respective strains was processed the same way as the atmospheric CH$_4$ treatment with the difference that filter cultures had been exposed to approximately 1000 p.p.m.v. CH$_4$ in air for two weeks before harvest. Samples of lyophilized cells were lysed by sonication in 20 μL buffer containing 4 M urea, 2.5% sodium deoxycholate (SDC) and 100 mM triethylammonium bicarbonate (TEAB). Samples were sonicated for 25 cycles (1 min on, 30 s off) with maximum amplitude in a cup horn sonicator with a recirculating chiller (Cup horn: model 413C2, Qsonica, Newtown, Connecticut, USA; Sonicator: Fisherbrandtm FB705, Thermo Fisher Scientific; Recirculating chiller: model 4905 Qsonica). Then, disulfide bridges were reduced with 1,4-dithiothreitol (DTT) at a final concentration of 5 mM and incubation at 54 °C for 30 min. Cysteines were alkylated with 15 mM iodoacetamide (IAA) and incubated for 30 min at room temperature in dark. To remove excess IAA, DTT solution corresponding to a final concentration of 5 mM was added. Calcium chloride solution (final concentration of 1 mM) and 1 μg Lysyl Endopeptidase (125-05061, FUJIFILM Wako Chemicals Europe, Neuss, Germany) were added to the samples and incubated for 5 h under gentle agitation at 37 °C for enzymatic digestion. After, samples were diluted with a buffer containing 100 mM triethylammonium bicarbonate (TEAB) and 1 mM CaCl$_2$ to lower the urea and sodium deoxycholate (SDC) concentration to 1 M and 0.65% v/v, respectively, resulting in a final sample volume of 80 μL. For digestion, 2 μg trypsin (V511A, Promega, Wisconsin, USA) were added and the samples incubated on a gently agitated shaker at 37 °C for 16 h. After digestion, SDC was precipitated by adding 50% formic acid to the sample (final concentration of 2.5% v/v). Samples were then incubated for 10 min and centrifuged at 16200 g for 15 min. Supernatants containing peptides were transferred to low-protein-binding tubes. The peptides were concentrated and cleaned up using DPX C18 pipette tips (DPX Technologies, XTR tips 10 mg C18AQ 300 Å) on a Tecan Fluent pipetting robot (Tecan Group Ltd., Männedorf, Switzerland). Purified peptide samples were dried in a vacuum concentrator and dissolved in 12 μL 0.1% formic acid. Peptide concentrations were measured on a spectrophotometer (Nanodrop ONE, Thermo Fisher Scientific) at 205 nm. 0.25 μg peptides per sample were loaded onto a liquid chromatograph (EASY-nLC1200, Thermo Fisher Scientific) equipped with an EASY-Spray column (C18, 2 μm, 100 Å, 50 μm, 50 cm). Peptides were fractionated using a 5–80% acetonitrile gradient in 0.1% formic acid over 120 min at a flow rate of 300 nL min$^{-1}$. The separated peptides were analyzed using a mass spectrometer (Orbitrap Exploris 480, Thermo Fisher Scientific). Data was collected in data dependent mode using a Top40 method. Annotated genomes of the three strains downloaded from MicroScope[72] served as databases for the CHIMERYS-based data search using Proteome Discoverer 3.0. Normalized abundances (scaling mode: On All Average) of proteins were further processed via Perseus[73]. Normal distribution of Log2 fold transformed data was visually screened using histograms. Proteins were filtered using a threshold of at least three valid values in at least one treatment ($n = 4$ per treatment). Proteins that passed the filtering were included in downstream analyses. Missing values were imputed from normal distribution using default settings (width = 0.3, down shift = 1.8). The Pearson correlation coefficient ranged from $\rho = 0.918$ to $\rho = 0.989$ between replicates and from 0.68 to 0.799 between replicates of the different treatments. Imputed protein abundance results of the 1.9 p.p.m.v. and 1000 p.p.m.v. treatments were used for

correspondence analysis (CA). CA was conducted in R[74] using the "ca" function of R package "ca"[75] version 0.71.1. The top 10% of proteins that contributed most to the inertia of the CA's first dimension were extracted using the function "get_ca_row" of the R package "factoextra"[76] version 1.0.7. To map the abundances and the hierarchical EggNOG[43] annotations of the top 10% proteins, the EggNOG annotations of the three strains were downloaded from MicroScope. For the in-depth analysis of trace gas oxidation, carbon assimilation, and the electron transport chain, the differences between treatments were tested using a two-sided $t$ test (s0 = 2) and permutation-based false discovery rate (FDR = 0.01). From the z-score normalized results, the proteins involved in trace gas oxidation, carbon assimilation via the serine cycle, and the electron transport chain were selected for further analysis using Python 3[68]. The core carbon and energy metabolism of *M. rosea* SV97 and *M. palsarum* NE2 were reconstructed manually using MicroScope annotations, the published metabolism entries of *M. gorgona* MG08[33], KEGG[77], and protein BLAST searches[78]. To conduct a thorough screening for putative carbon monoxide dehydrogenase subunits a blast-searchable database was constructed out of all sequences retrieved from the NCBI Identical Protein Groups resource[46] (accessed May, 30th 2023) using the search term carbon monoxide dehydrogenase (54076 amino acid sequence entries). Entries with an amino acid sequence length <30 were removed from the database prior to the blastp search. The annotated genomes of *M. rosea* SV97 and *M. palsarum* NE2, downloaded from MicroScope, were used as blastp queries (default settings, -outfmt 6 including slen). After a pre-filtering removing hits with an E-value > 1 and an alignment length <50% of the aligned database entry, the blastp output tables were manually evaluated (Supplementary data file, Dataset 10−11).

### Specific affinity

Filter cultures of *M. gorgona* MG08 and *M. palsarum* NE2 were pre-incubated on 10x diluted, EDTA-free NMS medium (DSMZ medium 921 with 10x the iron concentration and 1 µM lanthanum) for five months with air as sole energy and carbon source. The filter cultures were transferred into 250 mL glass bottles containing 250 mL of the mentioned medium. The glass bottles were capped with Safety-Caps (JR-S-11011, VICI AG International). For each strain, the CH$_4$ concentrations in the 50 mL headspace were adjusted to approximately 1.9, 14, 30, 70, and 175 p.p.m.v. by adding 0−2.5 mL 1000 or 5000 p.p.m.v. CH$_4$ in N$_2$ (HiQ, Linde, Sweden). The headspace pressure of all bottles was adjusted to approximately 1.1 bar using air. After, the strains were incubated for 48 h at 20 °C. The change in CH$_4$ was measured at incubation start, after 24 hours and 48 h as described above. Standard curves were created using 2.5 and 50 p.p.m.v. CH$_4$ in N$_2$ (HiQ, AGA, Sweden). At the end of the oxidation experiment, cells in filter cultures were quantified as described above. The change in mass of CH$_4$ in the headspace was calculated by applying the ideal gas law and adjusted for changes in pressure caused by gas removal during measurement. The mass of dissolved CH$_4$ at different CH$_4$ partial pressures in the headspace was calculated by applying the Henry's law solubility constant for CH$_4$ at 20 °C. The Michaelis-Menten CH$_4$ oxidation kinetics were modeled using R version 4.2.2 and the "nls" function of the "nlstools" R package[79] version 2.0.0, specifying the "michaelis" model and providing start values for $K_{m(app)}$ and $V_{max(app)}$. The $a_A^0$ was calculated by dividing $V_{max}$ by $K_m$.

### Growth on nitrogen from air

Filter cultures of *M. gorgona* MG08, *M. palsarum* NE2, *M. rosea* SV97, and *M. aurea* KYG were pre-incubated on 10x diluted, EDTA-free, and potassium nitrate (KNO$_3$)-free NMS medium (DSMZ medium 921 with 10x the iron concentration and 1 µM lanthanum) with air as sole energy, carbon, and nitrogen source. After three months, colony formation was checked via microscopy[33]. After 12 months, the activity of *M. gorgona* MG08 was measured as described in the section called Trace

gas oxidation experiments, with the only difference that the KNO$_3$-free medium was used.

For the detection of $^{15}$N$_2$ fixation via NanoSIMS, filter cultures were pre-incubated on 10x diluted, EDTA-free, and potassium nitrate (KNO$_3$)-free NMS medium with air as sole energy, carbon source, and nitrogen source. After three months, the filters were transferred into 250 ml glass bottles containing 50 ml of the KNO$_3$-free NMS medium. The bottles were capped with Safety-Caps (JR-S-11011, VICI AG International) and the headspace of each bottle was flushed for five minutes with compressed air using a gassing manifold. Afterwards, 50 ml of 98+ at% $^{15}$N$_2$ gas (NLM-363-1-LB, Cambridge Isotopes Laboratories, Tweksbury, Massachusetts, USA) were added using a gas syringe so that the $^{15}$N-N$_2$ in headspace atmosphere amounted approximately 23 at% of the total N$_2$. The headspace pressure was adjusted to 1.05 bar and 0.5 mL 1000 p.p.m.v. CH$_4$ in N$_2$ (HiQ, AGA, Sweden), 1 mL 1000 p.p.m.v. H$_2$ in N$_2$ (HiQ, AGA, Sweden), and 1 mL 1000 p.p.m.v. CO in N$_2$ (HiQ, AGA, Sweden) were added to the headspace using a gas syringe. To account for the natural abundance of $^{15}$N-N$_2$, a control without the addition of $^{15}$N$_2$ gas was prepared accordingly. The filter cultures were incubated at 20 °C for two months. The headspace concentration of CH$_4$, H$_2$, and CO during incubation was measured once a week as described in the section Trace gas oxidation experiments section and replenished if the concentration of CH$_4$, H$_2$, and CO dropped below 1.9 p.p.m.v., 0.5 p.p.m.v., 0.2 p.p.m.v., respectively.

After two months, the cells of the filter cultures were fixed by incubating the filters on 150 µl formaldehyde in 1×PBS (4% w/v) for an hour at room temperature. The cells were washed twice by letting the filters float on water for 5 min. The fixed cells were harvested as described in the section called Cell quantification, lyophilized, and stored at −80 °C until further processing. For NanoSIMS analysis the lyophilized cells were resuspended in 20 µl MilliQ water. 10 µl of the cell suspensions were deposited on antimony-doped silicon wafer platelets (7.1 × 7.1 × 0.75 mm, Active Business Company, Germany) and dried in air. The following two samples were prepared accordingly for NanoSIMS analysis: (i) The unlabeled *M. gorgona* MG08 cells, serving as control for the natural abundance of $^{15}$N$_2$; and (ii) the *M. gorgona* MG08 cells incubated with $^{15}$N$_2$ gas and trace concentrations of CH$_4$, H$_2$, and CO that were analyzed to see whether the strain is capable of fixing atmospheric nitrogen during growth with air as sole energy and carbon source.

The NanoSIMS measurements were performed on a NanoSIMS 50 l (Cameca, Gennevilliers, France) at the Large-Instrument Facility for Advanced Isotope Research at the University of Vienna. Before the data acquisition, analysis areas were preconditioned in situ by rastering of a high-intensity, defocused Cs$^+$ ion beam in the following sequence of high and extreme low ion impact energies (HE/16 keV and EXLIE/50 eV, respectively): HE at 25 pA beam current to a Cs+ fluence of 5.0E14 ions cm$^{-2}$; EXLIE at 400 pA beam current to a fluence of 5.0E16 ions cm$^{-2}$; and HE at 25 pA to a fluence of 5.0E14 ions cm$^{-2}$. Data were acquired as multilayer image stacks by repeated scanning of a finely focused Cs$^+$ primary ion beam (c. 80 nm probe size at approx. 2 pA beam current) over areas between 34 × 34 and 72 × 72 µm$^2$ at 512 × 512-pixel image resolution and a primary ion beam dwell time of 5 ms pixel$^{-1}$.

NanoSIMS images were generated and analyzed with the Open-MIMS plugin[80] in the image processing package Fiji[81]. All images were auto-tracked for compensation of primary ion beam and/or sample stage drift, and secondary ion signal intensities were corrected for detector dead-time and quasi-simultaneous arrival (QSA) of secondary ions, utilizing sensitivity factors ('beta' values) of 1.06, and 1.05 for C$_2^-$, and CN$^-$ ions, respectively. Regions of interest (ROIs) were defined in the $^{12}$C$^-$-ion image where each ROI corresponded to an individual cell. Cells touching the border of the image were omitted from the selection.

Acquisition cycles of all three analysis areas were reduced to 27 each to improve comparability among the measurements. Regions of interest were analyzed for their $^{15}N$ content by calculating the average value across acquisition cycles per analysis area, referred to by the $^{15}N/(^{14}N + ^{15}N)$ isotope fraction designated as at% $^{15}N$, which was calculated from the $^{12}CN^-$ signal intensities via:

$$at\%\ ^{15}N = \frac{^{12}C^{15}N^-}{^{12}C^{15}N^- + ^{12}C^{14}N^-} \qquad (4)$$

The natural abundance of $^{15}N$ in cellular biomass was inferred from unlabeled cells yielding $0.369 \pm 0.043$ at% (mean ± 1 SD).

## Plotting
Plots were created using Python 3[68] version 3.9.13 and R version 4.2.2. Python 3 packages used: Pandas[69] version 1.4.4, Matplotlib[70] version 3.5.2, and Seaborn[71] version 0.11.2. R packages used: ggplot2[82] version 3.2.0. Figures were finalized using Adobe Illustrator 2023.

## Reporting summary
Further information on research design is available in the Nature Portfolio Reporting Summary linked to this article.

## Data availability
The proteomics data have been deposited to the ProteomeXchange Consortium via the PRIDE[83] partner repository under accession code PXD046190. The data generated in this study are provided in the Source Data and Supplementary Data file. Source data are provided with this paper.

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

## Acknowledgements
We thank Alena Didriksen from the University of Tromsø for her involvement in cell quantification, Toril Anne Grønset and Jack-Ansgar Bruun from PRiME for their contributions to the comparative proteomics analysis and Cornelia Rottensteiner, Judith Prommer, and Margarete Watzka from the University of Vienna for the carbon content analysis and $^{15}N_2$ gas purity test. This study was supported by the Research Council of Norway FRIPRO Young Researcher Grant 315129 ATT, Living on Air, and Tromsø Research Foundation starting grant project Cells in the Cold 17_SG_ATT ATT. Mass spectrometry-based proteomic analyses were performed by UiT Proteomics and Metabolomics Core Facility (PRiME). This facility is a member of the National Network of Advanced Proteomics Infrastructure (NAPI), which is funded by the Research Council of Norway INFRASTRUKTUR-program (project number: 295910).

## Author contributions
T.S., M.M.S. and A.T.T. conceived the study. T.S., A.G.H., J.B., H.S., A.Sch., B.R., E.J.T., O.S., and A.T.T. performed experiments. T.S., H.S., A.Sch., A.S., A.R. and A.T.T. analyzed data. T.S. and A.T.T. created the figures. T.S., H.S., A.Sch., B.R., A.R., A.S., M.P, M.M.S, and A.T.T. contributed to methods. T.S. and A.T.T. wrote the manuscript with inputs from all authors.

## Funding

## Competing interests
The authors declare no competing interests.
