## [Peer Review File NEW · Nature Communications]

Physiological Basis for Atmospheric Methane Oxidation and Methanotrophic Growth on AirEditorial Note: This manuscript has been previously reviewed at another journal that is not operating a transparent peer review scheme. This document only contains reviewer comments and rebuttal letters for versions considered at *Nature Communications*.

Reviewer #1 (Remarks to the Author):

I read the latest version of the manuscript and the accompanying rebuttal with keen interest. My primary concern from the previous revision pertains to the difficulty in computing the cell-specific energy requirements of trace gas-oxidizing microorganisms. Specifically, I raised questions about the assessment of the number of living cells on the membranes used to measure trace gas (TG) oxidation rates.

The challenge lies in accurately measuring this variable, and the authors addressed it by the monitoring of N₂ incorporation in biomass by NanoSIMS to justify their assumption that all cells were alive and fixed N₂ during the assay. While their argument is plausible, the estimates remain somewhat delicate due to the inherent difficulty in gauging the number of cells on the membrane (potential loss of cells during the washing steps and cell sorting), the indirect evidence of living cells supported by labeled N incorporated in cell biomass (TG oxidation rate measurements were decoupled from NanoSIMS experiments), and the potential utilization of other energy sources in the air.

I believe it is important to acknowledge these complexities in the manuscript section that compares the energy yield with previous estimates. That being said, I believe the work represents a significant progress towards a better understanding of atmospheric chemosynthesis.

Reviewer #2 (Remarks to the Author):

The revised manuscript from Schmider et al. investigates the ability of a number of methanotrophs to grow on air by utilising the trace gases CH₄, H₂, and CO as a source of energy, as well as nitrogen sources (N₂ and otherwise).

The authors have paid careful attention to the queries and concerns of myself and the other reviewers, and their revision of the manuscript and explanations in the response to reviewers have done much to strengthen the manuscript. In its current form the manuscript significantly expands on our understanding of bacterial trace gas oxidation and makes a strong case that methanotrophs are able to utilise air as their sole source of energy and nitrogen.

I especially appreciate the authors inclusion of a control experiment showing that *M. gorgona* MG08 is unable to grow when supplied with air which lacks trace quantities of CH₄, CO, H₂. In response to my comments the reviewers point out the difficulty of creating a contaminant free environment, that doesn't contain substrates in addition of atmospheric CH₄, CO, H₂, that may contribute to growth. I am sympathetic of the difficulty of these experiments. However, if the authors wish to state that these methanotrophs are growing on air, it is important that they demonstrate that the reduced trace gases in air are required as at least the main sources of energy for growth. Extraordinary claims require extraordinary evidence. In this case, I believe it is sufficient to show this for one of the 4 species they identify as able to live on air. But in future work, I would suggest that these controls should be included as standard.

Minor comments:

Pg. 6 line 127 - 'performed' not 'perfromed'

Reviewer #3 (Remarks to the Author):

I have reviewed the author's responses to the reviewers, including the comments and questions I submitted (Reviewer 3). In all cases, I think they have provided appropriate answers, or have made the necessary changes.

I thank the authors for this attention to detail.

REVIEWERS' COMMENTS

We would like to thank the reviewers for their constructive comments and suggestions throughout the review process that were of great help in improving the overall quality of the study.

Reviewer #1 (Remarks to the Author):

I read the latest version of the manuscript and the accompanying rebuttal with keen interest. My primary concern from the previous revision pertains to the difficulty in computing the cell-specific energy requirements of trace gas-oxidizing microorganisms. Specifically, I raised questions about the assessment of the number of living cells on the membranes used to measure trace gas (TG) oxidation rates.

The challenge lies in accurately measuring this variable, and the authors addressed it by the monitoring of N₂ incorporation in biomass by NanoSIMS to justify their assumption that all cells were alive and fixed N₂ during the assay. While their argument is plausible, the estimates remain somewhat delicate due to the inherent difficulty in gauging the number of cells on the membrane (potential loss of cells during the washing steps and cell sorting), the indirect evidence of living cells supported by labeled N incorporated in cell biomass (TG oxidation rate measurements were decoupled from NanoSIMS experiments), and the potential utilization of other energy sources in the air.

I believe it is important to acknowledge these complexities in the manuscript section that compares the energy yield with previous estimates. That being said, I believe the work represents a significant progress towards a better understanding of atmospheric chemosynthesis.

We have included the following sentence at the end of the relevant section to make readers aware of the complexities:

Line 204 - 206: "However, we acknowledge that differences in activity between individual cells and potential inaccuracies in the quantification of cells contributing to observed activities might have introduced a minor error to our estimations."

Reviewer #2 (Remarks to the Author):

The revised manuscript from Schmider et al. investigates the ability of a number of methanotrophs to grow on air by utilising the trace gases CH₄, H₂, and CO as a source of energy, as well as nitrogen sources (N₂ and otherwise).

The authors have paid careful attention to the queries and concerns of myself and the other reviewers, and their revision of the manuscript and explanations in the response to reviewers have done much to strengthen the manuscript. In its current form the manuscript significantly expands on our understanding of bacterial trace gas oxidation and makes a strong case that methanotrophs are able to

utilise air as their sole source of energy and nitrogen.

I especially appreciate the authors inclusion of a control experiment showing that *M. gorgona* MG08 is unable to grow when supplied with air which lacks trace quantities of CH₄, CO, H₂. In response to my comments the reviewers point out the difficulty of creating a contaminant free environment, that doesn't contain substrates in addition of atmospheric CH₄, CO, H₂, that may contribute to growth. I am sympathetic of the difficulty of these experiments. However, if the authors wish to state that these methanotrophs are growing on air, it is important that they demonstrate that the reduced trace gases in air are required as at least the main sources of energy for growth. Extraordinary claims require extraordinary evidence. In this case, I believe it is sufficient to show this for one of the 4 species they identify as able to live on air. But in future work, I would suggest that these controls should be included as standard.

Minor comments:

Pg. 6 line 127 - 'performed' not 'perfromed'

The word has been changed accordingly.

Reviewer #3 (Remarks to the Author):

I have reviewed the author's responses to the reviewers, including the comments and questions I submitted (Reviewer 3). In all cases, I think they have provided appropriate answers, or have made the necessary changes.

I thank the authors for this attention to detail.